# SIRPα - CD47 axis regulates dendritic cell-T cell interactions and TCR activation during T cell priming in spleen

Anu Autio[1,☺,¤a], Huan Wang[1,☺], Francisco Velázquez[1,¤b], Gail Newton[1], Charles A. Parkos[2], Pablo Engel[3,4], Daniel Engelbertsen[1,¤c], Andrew H. Lichtman[1], Francis W. Luscinskas[1]*

**1** Center for Excellence in Vascular Biology, Department of Pathology, Brigham and Women's Hospital and Harvard Medical School, Boston, MA, United States of America, **2** Department of Pathology, University of Michigan Medical School, Ann Arbor, Michigan, United States of America, **3** Immunology Unit, Department of Biomedical Sciences, Medical School, University of Barcelona, Barcelona, Spain, **4** Institut d'Investigacions Biomèdiques August Pi i Sunyer, Barcelona, Spain

☺ These authors contributed equally to this work.
¤a Current address: MediCity Research Laboratory, University of Turku, Turku, Finland
¤b Current address: Repertoire Immune Medicines, Cambridge, MA, United States of America
¤c Current address: Department of Clinical Science, Skåne University Hospital, Lund University, Malmo, Sweden
* fluscinskas@partners.org

**Data Availability Statement:** All relevant data are within the paper and its Supporting Information files.

## Abstract

The SIRPα-CD47 axis plays an important role in T cell recruitment to sites of immune reaction and inflammation but its role in T cell antigen priming is incompletely understood. Employing OTII TCR transgenic mice bred to *Cd47*-/- (*Cd47*KO) or SKI mice, a knock-in transgenic animal expressing non-signaling cytoplasmic-truncated SIRPα, we investigated how the SIRPα-CD47 axis contributes to antigen priming. Here we show that adoptive transfer of *Cd47*KO or SKI Ova-specific CD4+ T cells (OTII) into *Cd47*KO and SKI recipients, followed by Ova immunization, elicited reduced T cell division and proliferation indices, increased apoptosis, and reduced expansion compared to transfer into WT mice. We confirmed prior reports that splenic T cell zone, CD4+ conventional dendritic cells (cDCs) and CD4+ T cell numbers were reduced in *Cd47*KO and SKI mice. We report that in vitro derived DCs from *Cd47*KO and SKI mice exhibited impaired migration in vivo and exhibited reduced CD11c+ DC proximity to OTII T cells in T cell zones after Ag immunization, which correlates with reduced TCR activation in transferred OTII T cells. These findings suggest that reduced numbers of CD4+ cDCs and their impaired migration contributes to reduced T cell-DC proximity in splenic T cell zone and reduced T cell TCR activation, cell division and proliferation, and indirectly increased T cell apoptosis.

## Introduction

Signal regulatory protein α (SIRPα, CD172a) is a type I transmembrane receptor glycoprotein containing an intracellular domain with two immunoreceptor tyrosine-based inhibitory

**Funding:** This study was supported by the National Institutes of Health grant R01 HL125780 awarded to FWL (www.nhlbi.nih.gov). The funders had no role in study design, data collection and analysis, decision to publish, or preparation of the manuscript.

**Competing interests:** The authors declare no competing financial interests.

motifs (ITIMs) that associate with Src homology region 2 (SH2)-containing protein tyrosine phosphatase 1 (SHP-1) and SHP-2 (reviewed in [1]). SIRPα is highly expressed in myeloid leukocytes but not T cells [2, 3]. CD47 is a broadly expressed type III transmembrane glycoprotein that interacts in the same plasma membrane (in cis) with integrins, and in trans with SIRPα and SIRPγ [3–7]. Binding of CD47 on circulating blood cells to SIRPα expressed on splenic myeloid cells transmits a negative "don't eat me" inhibitory signal, preventing the phagocytosis of circulating hematopoietic cells [8].

The current understanding of the contribution of the SIRPα-CD47 axis to adaptive immunity is incomplete, especially regarding immune cell activation, proliferation, and trafficking. The importance of SIRPα - CD47 signaling pathway was previously demonstrated in mice genetically deficient in *Cd47* (*Cd47*KO) or in mice that express a mutant, non-signaling cytoplasmic truncated *Sirpa* gene (SKI). Loss of this signaling pathway was protective in inflammatory disease models including experimental autoimmune encephalomyelitis (EAE), inflammatory bowel disease, ischemia/reperfusion injury and diabetes (reviewed by [4, 8]) with a few exceptions [9, 10]. However, the underlying mechanisms remain incompletely defined.

The SIRPα-CD47 axis plays an important role in leukocyte migration, a crucial step for innate and adaptive immunity. For example, SIRPα has been shown to regulate myeloid leukocyte migration in vivo and in vitro [4, 11–14]. Similarly, we have shown that CD47 regulates leukocyte trafficking through its "*in cis*" association with the leukocyte β2 integrins LFA-1 (CD11a/CD18) and Mac-1 (CD11b/CD18), and with VLA-4 (CD49d/CD29) integrins, and through its important role in endothelial-dependent and epithelial-dependent signaling during leukocyte diapedesis in vivo and in vitro [5, 6, 15–20].

Previous reports have shown that the number of splenic CD4+ classic DCs (cDC) were reduced in *Cd47*KO and SKI mice compared to WT mice [9, 13, 21–23]. In addition, the absolute number of splenic CD4+ T cells and fibroblastic reticular cells (FRC), and the size of the splenic T cell zone and the fibroblastic reticular network were reduced in these mice. Lack of FRC impaired the production of CCL19 and CCL-21, which are chemoattracts for both CD4+ T cells and CD4+ cDCs, as well as loss of the T cell autocrine IL-7. These defects were traced to a loss in production of TNF receptor ligands lymphotoxins LTα3 and TNF-α by SIRPα+ CD4+ cDCs [24]. These findings were recently attributed to a SIRPα deficiency specifically in CD11c+ immune cells [24–27], which affect T cell priming [27]. Because of the importance of the SIRPα-CD47 axis in immune cell homeostasis within the spleen, our goal was to identify other events in antigen priming contributed by the SIRPα - CD47 adhesion pathway, beyond the reductions in immune cell numbers. Here we used OTII transgenic mice bred to *Cd47*KO and SKI mice, and flow cytometry quantification together with 4-color immunofluorescence microscopy of splenic tissues for adoptive transfer studies. We suggest that the dramatic reduction in CD4+ cDC numbers and their impaired migration can explain the observed reduction in T cell proximity to cDC in splenic T cell zone upon Ag immunization, which in turn, can account for the reduced T cell TCR activation, cell division and proliferation, and indirectly increased apoptosis.

## Materials and methods

### Mice

C57BL/6 wild type (WT) mice from the Jackson Laboratories (Bar Harbor, ME, USA) were used to establish a breeding colony. OT-II T cell receptor transgenic (Tg) mice (*Tg(TcraTcrb)425Cbn*) and *Cd47*KO mice [28] were purchased from Jackson Laboratory (Bar Harbor, ME USA) and used to establish breeding colonies. The *Sirpatm1Nog* mouse, abbreviated here by

SKI, was reported by Inagaki and colleagues [29], and were obtained from Dr. Benjamin Neel while he was a faculty member at the University of Toronto, Ontario, Canada. These authors showed that the mutant SIRPα protein does not undergo tyrosine phosphorylation or form a complex with SHP-1 or SHP-2, and hence does not transmit intracellular signals. *Cd47*KO and SKI mice were back bred onto C57BL/6 mice for > 10 generations. OT-II mice were bred to *Cd47*KO and SKI mice to generate OT-II:*Cd47*KO (KO:OTII) and OT-II SKI (SKI:OTII) mice. Both male and female mice were used between 8–14 weeks of age. All animals used in this study were bred and/or housed in the pathogen-free animal facility and used in accordance with Institutional Animal Care and Use Committee guidelines of Mass General Brigham. Human studies have been reviewed and approved by Mass General Brigham Institutional Review Board and in accord with the ethical standards of Helsinki Declaration of 1975 (revised 2008).

## Materials

Recombinant human and mouse cytokines were obtained from PeproTech, Inc (Rocky Hill, NJ USA). RPMI1640, DPBS without $Ca^{2+}$ and $Mg^{2+}$ (DBPS-), PBS with or without $Ca^{2+}$ and $Mg^{2+}$ (PBS-) and DMEM were purchased from Life Technologies (Carlsbad, CA USA). Anti-human LFA-1 monoclonal antibody (mAb) TS1/22 hybridoma was obtained from the ATCC (Manassas, VA USA) and used as purified IgG. Anti-human ICAM-1 Ab (clone Hu5/3) was used as purified IgG [30] and anti-human SIRPα mAb (clone SAF17.2) [31] was used as purified IgG. Murine Miap410 (IgG1, clone BP0283) and isotype control MOPC-21 (BP0083) were purchased from BioXCell (Lebanon, NH USA). Miap410 Ab recognizes both human and mouse CD47 and specifically blocks CD47 binding to SIRPα but does not affect CD47-LFA-1 integrin "in cis" interactions [9]. Ovalbumin (OVA) and OVA$_{323-339}$ peptide were purchased from ThermoFisher (Waltham, MA USA). mAb APA1/1 reacts with the activated intracellular domain of the epsilon chain of the CD3 molecule [32]. Exposure of the epitope precedes CD3 phosphorylation and recruitment and activation of ZAP70, which initiates the signaling cascade produced by T cell activation. Murine CD4$^+$ T cell isolation kits were obtained from Miltenyi Biotec (Auburn, CA USA). CFSE, CellTrace Violet and CellTrace Far Red Cell Proliferation Kits were from Life Technologies (Carlsbad, CA USA). Human Monocyte Enrichment and RosetteSep Human T Cell Enrichment Kits were purchased from StemCell Technologies (Tukwila, WA USA). LIVE/DEAD™ Fixable Far Red Dead Cell Stain Kit and Fluorescent APC-Annexin V apoptosis detection kits were purchased from Life Technologies (Waltham, MA USA).

## Murine cell isolation

CD4$^+$ T cells were purified from pooled spleens and LNs by Miltenyi Biotec immunobead separation and were > 93% by flow cytometry [9, 15]. CD11c$^+$ DCs were generated from freshly isolated murine bone marrow cells by culture for 7–10 d in RPMI1640 supplemented with 10% FBS, GlutMAX, mIL-4 (20 ng/mL), mGM-CSF (50 ng/mL), penicillin and streptomycin (P/S). Bone marrow generated DCs (BMDC) were > 60% CD11c$^+$ as determined by flow cytometry with similar recoveries for each strain. Total cell counts were determined by a Coulter Z2 series particle analyzer (Beckman Coulter, Indianapolis, IN, USA).

## In vitro human monocyte derived dendritic cells (MDDC) and CD3$^+$ T cells

Human peripheral blood monocytes were isolated from leukocyte reduction collars obtained from volunteer donors through the Harvard Crimson Cell core facility (Boston, MA USA) and stored in liquid nitrogen as frozen stocks in FBS-10% (v/v) DMSO. To generate MDDC, frozen

monocytes were thawed and washed once, and cultured in RPMI1640-10% FBS, 1 mM Gluta-MAX, P/S, hIL-4 (20 ng/mL) and hGM-CSF (20 ng/mL) for 7–10 days. Resulting MDDCs were > 60% CD11c[+] by flow cytometry. Human CD3[+] T cells were isolated by negative selection (STEM Cell Technologies, Vancouver, British Columbia, Canada) from sodium citrate anti-coagulated whole blood obtained from healthy human volunteer donors under an approved Brigham and Women's Institutional IRB protocol. T cells routinely were >95% CD3[+] purity and used immediately or rested overnight before use.

## Flow cytometry analysis

Single cell suspensions of immune cells were analyzed by flow cytometry using optimized concentrations of mAbs to: CD3 (clone 17A2), CD4 (RM4-4), CD8 (53–6.7), CD47 (Miap301) Class II (I-Ab) (AF6-120.1), CD44 (IM7), CD62L (MEL14), CD86 (GL-1), CD11b (M1/70), CD11c (N418), CD25 (PC61), SIRPα (P84), LFA-1 (M17/4) and VLA-4 (9C10), and mAb B20.1-PE and MR9-4-FITC (Biolegend, San Diego, CA USA) that recognize the OTII antigen. PI or Far Red viability dye was used to exclude all non-viable cells from the analysis. mAb to CD16 and CD32 were included to block non-specific mAb binding to FcγRs. FACS analysis was performed on a BD FACSCanto II flow cytometer and the data analyzed by FlowJo software (FlowJo LLC, Ashland, OR).

## Analysis of CD4[+] T cell apoptosis

Splenocytes (1–3 x10[6]) were stained using an Annexin V-APC staining kit following the manufacturer's instructions. OTII T cells were identified by staining using the two mAbs, B20.1-PE and MR9-4-FITC, that detect the OTII TCR, and cell fluorescent analyzed by standard flow cytometry.

## Adoptive transfer studies

CD4[+] T cells were labeled with 5 μM CellTrace Violet dye following manufacturer's instructions. 4.0×10[6] labeled T cells in 200 μL of PBS- were injected i.p. or i.v. per mice. 24 h later, mice were injected i.p. with 100 μg of OVA in 100 μL of PBS- or PBS- alone. 72 h after T cell transfer, spleens were harvested, and single cell suspensions were analyzed for surface antigens and dye dilution by flow cytometry [15].

## OTII T cell proliferation indices

The recovery of OTII T cells from Ova immunized mice was calculated as the number of live, dye labelled CD4[+] cells normalized to the total CD4[+] cell count in the spleen. The division index is the average number of cell divisions that a cell in the original population has undergone and the calculation includes the peak of undivided T cells using FlowJo software. The proliferation index is the total number of divisions divided by the number of cells that entered division using FlowJo software. The proliferation index only includes cells that underwent at least one division and only responding cells are included. Our initial studies showed i.v. and i.p. infusion of labeled T cells in WT recipient mice resulted in same amount of T cell expansion after antigen (Ag) in WT mice, thus subsequent studies used i.p. injection. Leukocyte number was determined by Beckman Coulter Z1 D Particle Counter and corrected for viability by PI viability stain and flow cytometry.

## Adoptive transfer of BMDC

BMDC were labeled with one of three fluorescence dyes, CellTrace Violet, CFSE or CellTrace Far Red, following the manufacturer's instructions. Labeled BMDC were counted, resuspended in a 1:1:1 ratio to a final number of $3–5 \times 10^6$ cells per 30 μL of PBS, and the ratio of pre-injection sample was determined by flow cytometry. Recipient mice received a single 30μL cell sample injection into the hock of the hind leg and inguinal LNs were harvested 40 h later. The total number of viable CD11c$^+$ cells was determined by cell counting, and the ratio of 3-color fluorescence determined by flow cytometry. The recovery of migrated DCs was calculated as follows: number of CD11c$^+$ of each strain's cell type recovered in LN ÷ number of each strain's DCs injected. Data were expressed as DC migration (% recovery of labeled live DCs).

## In vivo transmigration assay (subcutaneous air pouch model)

Air pouches were created in the dorsal portion of the back of 8–10 week old C57BL/6 WT and SKI male mice [16]. PBS or PBS containing murine recombinant TNF-α (500ng/100μL) was injected in each air pouch, and 4 or 24 h later cell infiltrates were harvested by lavage with repeated PBS washes. The recovered volume was measured and the number of recovered cells was determined by hemocytometer. The frequency of CD3$^+$ T-cells, neutrophils and monocyte/macrophages were determined by staining with primary labeled mAb specific for these cell types and acquired on a FACScalibur flow cytometer (BD Biosciences). FACS data were analyzed using FlowJo software (Tree Star Inc., Ashland, OR).

## In vivo detection of TCR activation

mAb APA1/1 recognizes a conformational change in the CD3 epsilon protein of the TCR complex that occurs shortly after the TCR fully engages stimulatory peptide-MHCII ligands in vivo and in vitro [32, 33]. CFSE-labeled CD4$^+$ OTII cells ($8 \times 10^6$) were i.p. injected into mice. After 60-h mice were immunized i.p. with 100 μL of PBS- containing 100 μg OVA or PBS-alone. Based on pilot studies we found that 6 h post OVA immunization provided optimal binding of mAb APA1/1. Mice were euthanized, inguinal LNs and spleens harvested, single cell suspensions prepared, and cells subjected to intracellular staining with APA1/1-APC mAb followed by flow cytometric analysis.

## In vitro DC-T cell conjugate formation

In vitro derived human MDDCs were labeled with 1 μM CellTrace Violet and rested in culture for 48 hr. Freshly isolated human CD3$^+$ T cells were labeled with 5 μM CFSE, mixed in 1:2 ratio (DC:T cell) with MDDCs, centrifuged at 800 rpm for 1 min, and followed by a 2h incubation at 37˚C. HLA mismatch between donors of T cells and DCs led to conjugate formation. Cells were resuspended by vortex and analyzed immediately by flow cytometry. In some studies, function blocking mAb to LFA-1 (clone TS1/22), ICAM-1 (Hu5/3), SIRPα (SAF17.2) or CD47 (Miap410) or control IgG1 (MOPC2/1) were included each Ab at 20 μg/mL. For murine conjugate formation, BDDCs were labeled in PBS with 0.5 μM Violet dye, washed, and then cultured at 37˚C with Ova (1μg/ml) or PBS alone for 2 h. CFSE-labeled naïve OTII CD4$^+$ T cells ($1 \times 10^5$) and violet-labeled BMDCs ($1 \times 10^5$) were mixed in a 1:1 ratio with mAbs, centrifuged for 1 min at 800 rpm, and incubated for 2 h at 37˚C. Thereafter cells were vortexed and immediately analyzed by two-color flow cytometry. Conjugates of T cell—APCs were located within the Violet-CFSE double positive quadrant as determined by FlowJo software.

### Epifluorescence microscopy and image acquisition of splenic T cell zone

CFSE-labeled WT OTII T cells ($4x10^6$) were i.v. injected into mice, and 24-h later spleens were harvested, immediately imbedded in OCT, snap frozen, and sectioned using a Leica CM3050C cryostat at 5 μm intervals. Cut sections were fixed in 4% PFA and stained with primary labeled mAb as detailed previously [9]. T cell zones and the number of transferred OTII T cells within the T cell zones in splenic sections were determined by four-color fluorescence microscopy: T cell zones were defined by CD11c stained DCs (red), contained CFSE labeled OTII T cells (green), and were bordered by B cell follicles (blue) and the marginal zone macrophages (white) (see Fig 4C). Blinded analysis of the splenic T cell x-y circumferential area in $μm^2$ (red staining) and distances between CFSE$^+$ T cells and CD11c$^+$ DCs (See Fig 6 and S4 Fig. were measured using ImageJ software and IMARIS software (Oxford Instruments, Concord, MA).

### Fluorescence microscopy

Fluorescence micrographs of stained splenic sections were obtained with a Zeiss Axioplan2 epifluorescence microscope (Carl Zeiss, White Plains, NY) with Chroma single-channel ET filter sets, 10x and 20x Plan-Apochromat objectives (NA 0.45 and 0.8, respectively) and a Coolsnap HQ camera controlled by MetaMorph 7.8 (Molecular Devices, Sunnyvale, CA USA). To identify OTII T cell and splenic CD11c$^+$ DC locations in splenic sections, 10-μm-thick z-stacks were collected at 0.2-μm intervals, the contrast and brightness of each stained test and isotype control mAb were adjusted using Image J software in a blinded fashion. Representative compensated images were reconstructed into 3-dimension projections using MetaMorph software v7.8. Quantifying the distances in μm between transferred CFSE$^+$ T cells and endogenous CD11c$^+$ DCs in the T cell zones of splenic white pulp was performed using IMARIS software. IMARIS masking was applied to identify CFSE$^+$ T cells and to measure the distance to the nearest CD11c$^+$ DCs. Representative masked images for each strain of mice are presented in S4 Fig.

### Statistical analysis

Data are expressed as means ± SEM. One-way or two-way ANOVA followed by a Tukey post-test analysis were used for comparisons between three or more groups. Student's *t* Test with Welch's post-hoc correction was used for comparisons between two treatment groups using Prism software v8.4.2 (GraphPad Software, La Jolla, CA). $p< 0.05$ was considered statistically significant.

## Results

### CD47 and SIRPα are required for Ag-induced CD4$^+$ T cell expansion in vivo

To define the role(s) of CD47—SIRPα receptor-ligand pair in Ag priming of T cells, *Cd47*KO and SKI mice were bred to OTII Tg mice to generate KO:OTII and SKI:OTII Tg mice, respectively. To separately test the requirement for CD47 and SIRPα in a non-self Ag challenge, dye labeled KO:OTII T cells were adoptively transferred into *Cd47*KO recipients, SKI:OTII T cells were transferred into SKI recipient mice, and WT:OTII T cells were transferred into WT recipients as a positive control (Fig 1A). Twenty-four h after T cell transfer, the mice were immunized i.p. with Ova, and 72 h later splenocytes were recovered to assess OTII T cell responses.

The OTII T cell responses were much reduced in *Cd47*KO and SKI mice compared to WT mice even though the CD4$^+$ violet dye dilution plots demonstrated that the same number of

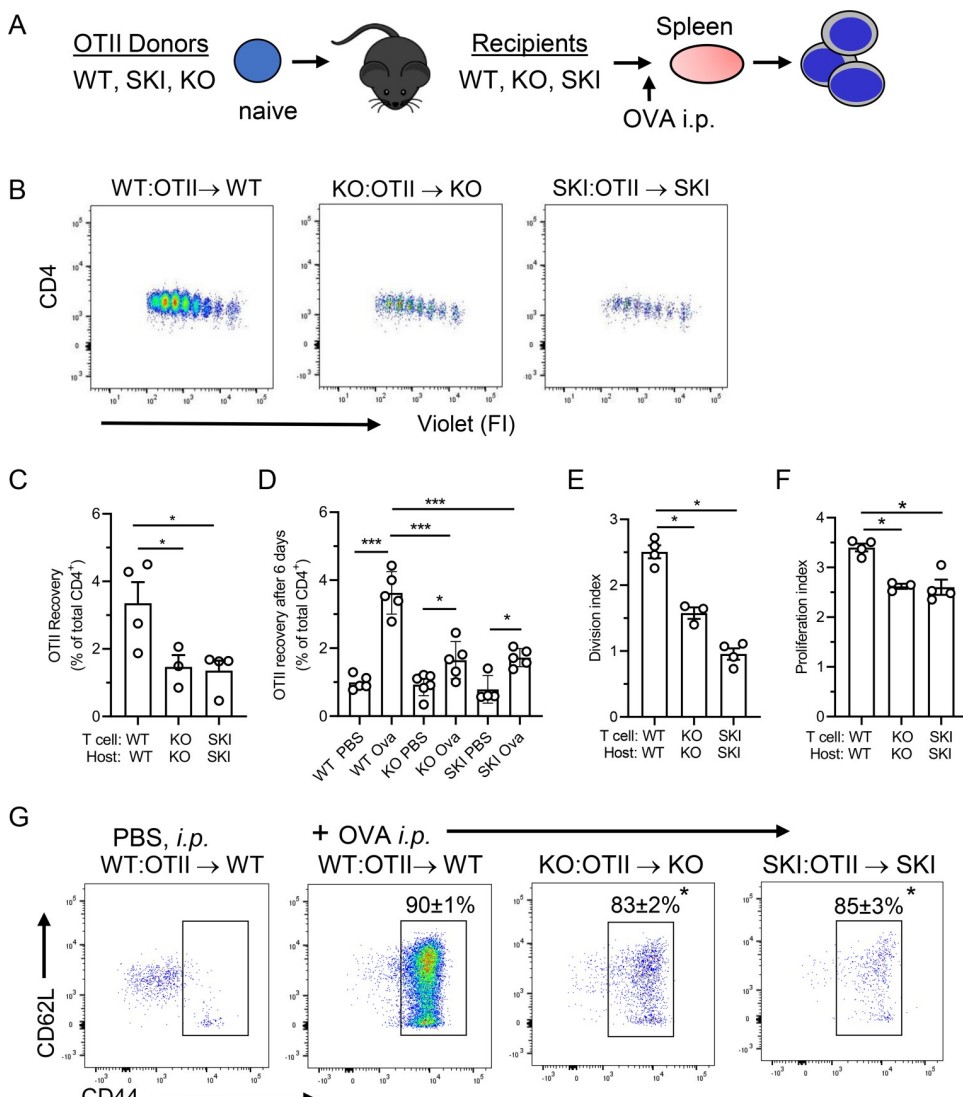

**Fig 1. Interaction of CD47 with its ligand SIRPα is necessary for Ag-induced CD4$^{+ T}$ cell expansion in vivo. (A)** T cell adoptive transfer scheme. **(B)** Proliferation of adoptively transferred violet dye labeled CD4$^+$ OTII T cells recovered from WT, *Cd47*KO or SKI recipient mice. Each peak indicates one cell division. **(C-F)** OTI cells recovered after 72 h or 6 days, and division and proliferation indices of transferred OTII T cells post OVA immunization. Means ± SEM, 4–6 animals per group, and representative of 2 independent experiments. * p<0.05, ***p<0.001 compared to WT by one-way ANOVA with Tukey test. **(G)** Representative CD62L and CD44 surface expression on gated double positive CD4$^+$ WT, *Cd47*KO and SKI OTII T cells recovered from spleens of SKI, *Cd47*KO or WT mice. The number above the open rectangle is the average percentage of CD44$^{hi}$ effector T cells. Data are means ± SEM from 3–4 mice per group. * p< 0.05 compared to WT-OVA by one-way ANOVA with Tukey test.

cell divisions had occurred (Fig 1B and 1C). The recovery of OTII T cells after 72 hours from *Cd47*KO (56% reduction) and SKI (60% reduction) recipient mice was significantly reduced even after the data were normalized to total splenic CD4$^+$ cell counts, which were reduced by similar amounts in *Cd47*KO and SKI mice as previously reported [26]. Notably, even after 6 days post-Ova immunization the responses of transferred OTII T cells in KO and SKI recipient mice did not catch up to those observed in WT mice (Fig 1D). Consistent with reduced recoveries, the *Cd47*KO and SKI mice showed deficiencies in both cell division and proliferation indices suggesting defects in both the T cell activation and expansion (Fig 1E and 1F). The

recovered OTII T cells did, however, display an activated phenotype of uniformly elevated expression of CD44, with a modestly lower percent of cells activated in SKI and *Cd47*KO mice (Fig 1G).

One explanation for limited proliferation of OTII cells in KO and SKI mice is reduced survival because previous reports found that blood cells in both KO and SKI mice either were phagocytosed and/or exhibited increased apoptosis [15, 34]. We therefore assessed phagocytosis of CFSE-labeled KO, SKI or WT OTII T cells by splenic phagocytic cells 24 h after transfer into KO, SKI or WT mice prior to immunization, and did not observe significant phagocytosis of transferred cells in any of the recipient mice. On the other hand, we did observe a 1.5-fold increase in apoptotic OTII T cells in KO and SKI mice 48 and 72 h after OVA immunization (S1A and S1B Fig). The PBS control showed that transferred OTII T cells were maintained at normal levels in all three strains until 72 h when the amount of apoptotic OTII T cells increased significantly in SKI ($p<0.05$) and *Cd47*KO ($p< 0.01$) compared to WT mice. This amount of apoptosis likely explains in part the poor responses after Ova immunization. Hence, we pursued the possibility of defects in both SKI and KO hosts by assessing whether these mice support Ag priming of transferred WT OTII T cells.

## SKI and *Cd47*KO mice support limited Ag induced expansion of naïve CD4+ T cells

Equal numbers of dye-labeled naïve WT OTII T cells were transferred into WT, *Cd47*KO and SKI recipients, the mice were immunized with OVA 24 h later, and splenocytes were harvested after 72 h for analysis as in Fig 1. The CD4+ dye-dilution results and the recovery of WT OTII T cells from *Cd47*KO and SKI recipients were significantly less than in WT recipients, and we observed a small but significant reduction in the percentage of activated CD44+ CD4+ OTII T cells (Fig 2A–2C). Both division and proliferation indices were reduced compared to WT (Fig 2D and 2E). These data indicate that *Cd47*KO and SKI hosts failed to support robust WT T cell activation and expansion as compared to WT recipients. As anticipated, adoptive transfer of labeled SKI OTII cells into KO or KO OTII cells into SKI mice, also resulted in reduced Ag dependent responses (Fig 2F–2I). We note that transfer of KO T cells into WT recipient leads to their rapid clearance from WT animals, as previously reported, hence this arm was not performed [35, 36].

The poor recovery of SKI OTII T cells in WT recipient mice was unexpected because SIRPα is minimally expressed by T cells while normal amounts of CD47, and LFA-1 and VLA-4 integrins were observed (S2A Fig). Further investigation revealed a defect in LFA-1 integrin dependent arrest on immobilized ICAM-1 under shear flow conditions in vitro (S2B Fig) and reduced recruitment of blood CD4+ T cells as well as neutrophils in a dermal air pouch model of TNF-α-induced inflammation (S2C–S2E Fig). These results suggest SKI OTII T cells have defects in LFA-1 function, which could explain in part the reduced Ag dependent response.

## CD47 and SIRPα are not required for Ag-induced CD4+ T cell–DC conjugate formation and T cell proliferation in vitro

The data presented thus far suggested that either the DCs in SKI and *Cd47*KO mice fail to activate OTII T cells, or that the recipient mice do not support survival of OTII T cells, or both. To address these points, we evaluated Ova-induced T cell-DC conjugate formation and T cell clonal expansion in vitro. We observed no reduction in conjugate formation between Ova peptide loaded *Cd47*KO and SKI bone marrow derived DCs (BMDC) and naïve WT, *Cd47*KO and SKI OTII T cells (Fig 3A). In addition to genetic knockout cells, function blocking mAb to ICAM-1, LFA-1, CD47 and SIRPα were utilized (Fig 3B). Conjugate formation was blocked to

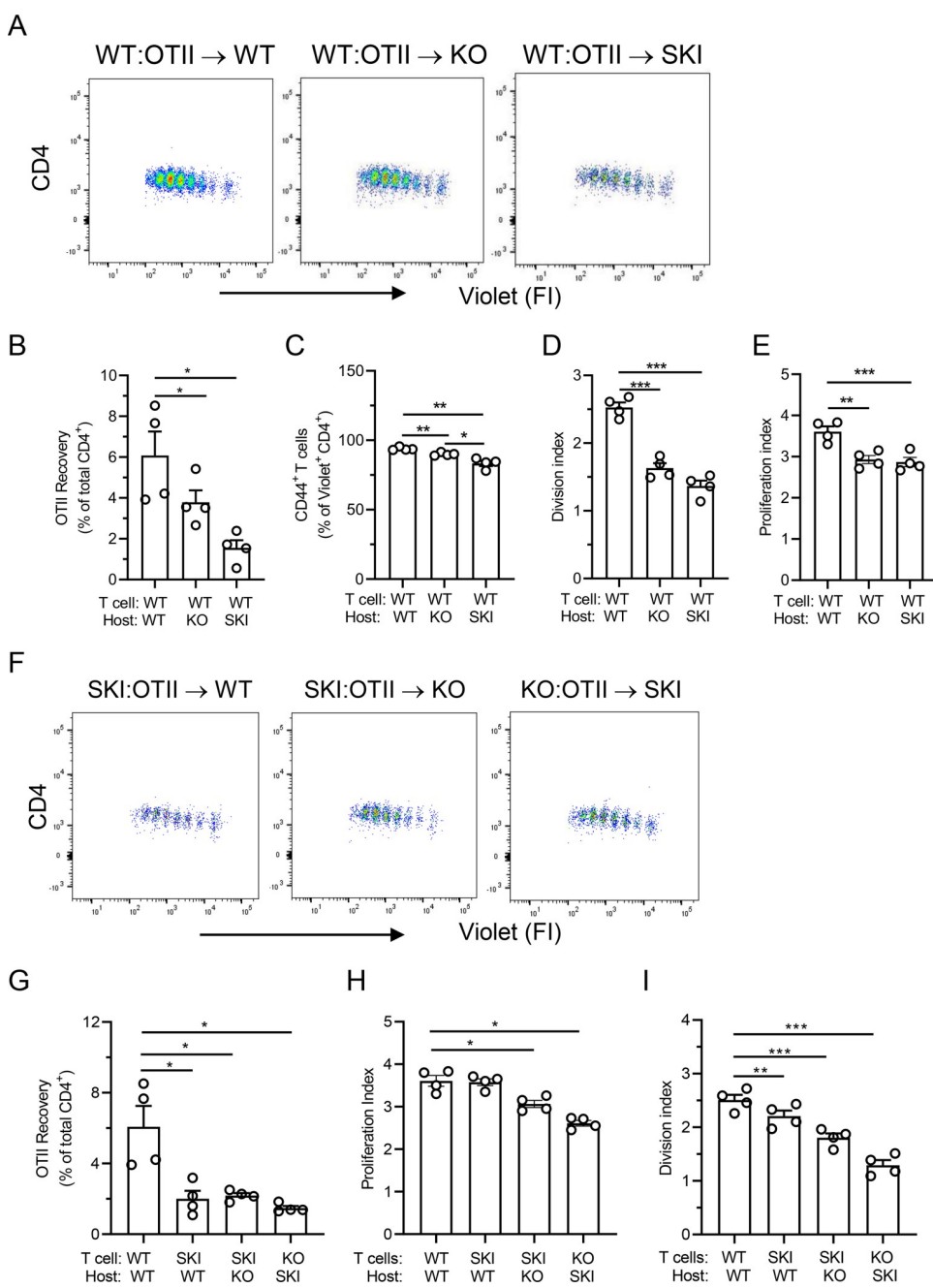

**Fig 2. CD47 deficiency or loss of SIRPα signaling in recipient mice results in severe reduction in CD4$^+$ T cell expansion in vivo. (A)** Proliferation of adoptively transferred naïve dye labeled CD4$^+$ OTII T cells recovered from WT, *Cd47*KO or SKI recipient mice. Means ± SEM from 3–4 animals per group from 2 separate experiments. **(B-E)** CD4$^+$ OTII T cell expansion in 72 h, % CD44$^{hi}$ activated T cells, and proliferation and division indexes relative to WT mice. Means ± SEM from 4 mice per group, results are representative of 2 independent experiments. $^*$ p<0.05, $^{**}$ p< 0.01, $^{***}$ p<0.001 versus WT mice by one-way ANOVA with Tukey test. **(F)** Proliferation of adoptively transferred naïve dye-labeled SKI or *Cd47*KO CD4$^+$ OTII T cells recovered from WT, *Cd47*KO or SKI recipient mice. Means ± SEM from 4 animals per group. **(G-I)** CD4$^+$ OTII T cell expansion in 72 h, % CD44$^{hi}$ activated T cells, and proliferation and division indexes relative to WT mice. Means ± SEM from 4 mice per group.

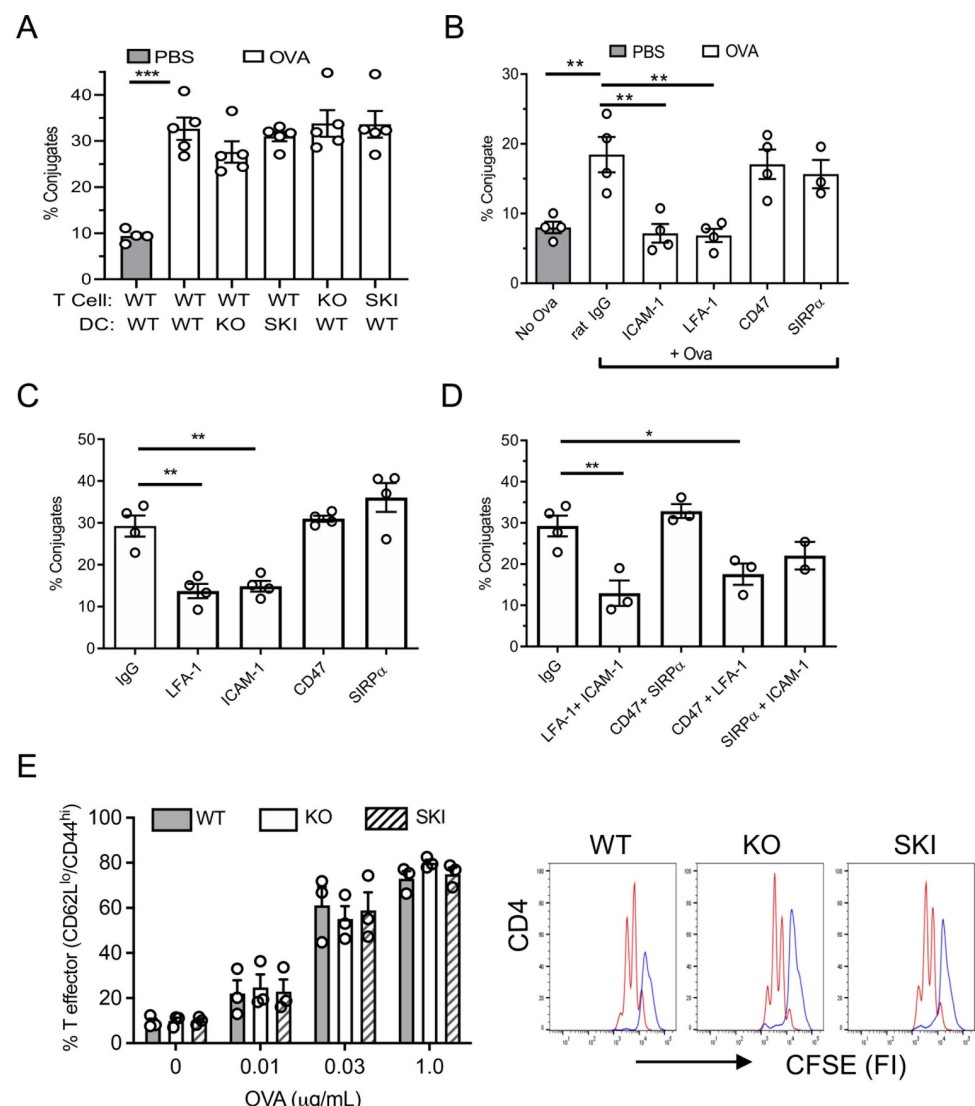

**Fig 3. Interaction of CD47 and SIRPα does not play a role in OVA-induced CD4⁺ T cell production in vitro. (A)** Conjugate formation between murine WT, *Cd47*KO and SKI OTII CD4⁺ T cells with BMDC from WT, SKI and *Cd47*KO mice at baseline and with OVA or PBS control. Means ± SEM from 3 independent studies, each performed with duplicate samples. *** p< 0.001, by one-way ANOVA with Tukey test. **(B)** Conjugate formation of WT BMDC and OTII CD4⁺ T cells in presence of various function blocking mAb specific for molecules indicated on x-axis. Means ± SEM from 3 independent studies, each performed with duplicate samples. ** p<0.01 by one-way ANOVA with Tukey test. **(C,D)** Conjugate formation between human CD4⁺ T cells and MDDC in presence of IgG control or blocking mAb specific for molecules indicated. Means ± SEM from n = 3 independent studies each performed with duplicate samples. * p<0.05, ** p<0.01, *** p< 0.001 by one-way ANOVA with Tukey test. **(E)** Analysis of WT, *Cd47*KO and SKI CD4⁺ T cell activation (CD44^hi CD62L^lo) and proliferation assessed by dye dilution after 48 h of coculture in vitro with indicated MDDCs, with or without OVA. Proliferation of *Cd47*KO and SKI was not significantly different from WT. PBS treated, purple line; Ova stimulated, red line. The results are representative of 3 independent experiments.

baseline by mAb to ICAM-1 or LFA-1, which is in line with the literature [37]. In contrast, function blocking anti-CD47 or SIRPα mAbs had no significant effect. We also verified these results in human cells in the conjugate assay, which was based on the HLA mismatch between blood monocyte derived DCs from one donor incubated with CD4⁺ T cells from different donors (Fig 3C and 3D). Function blocking mAb to CD47 or SIRPα alone or in combinations,

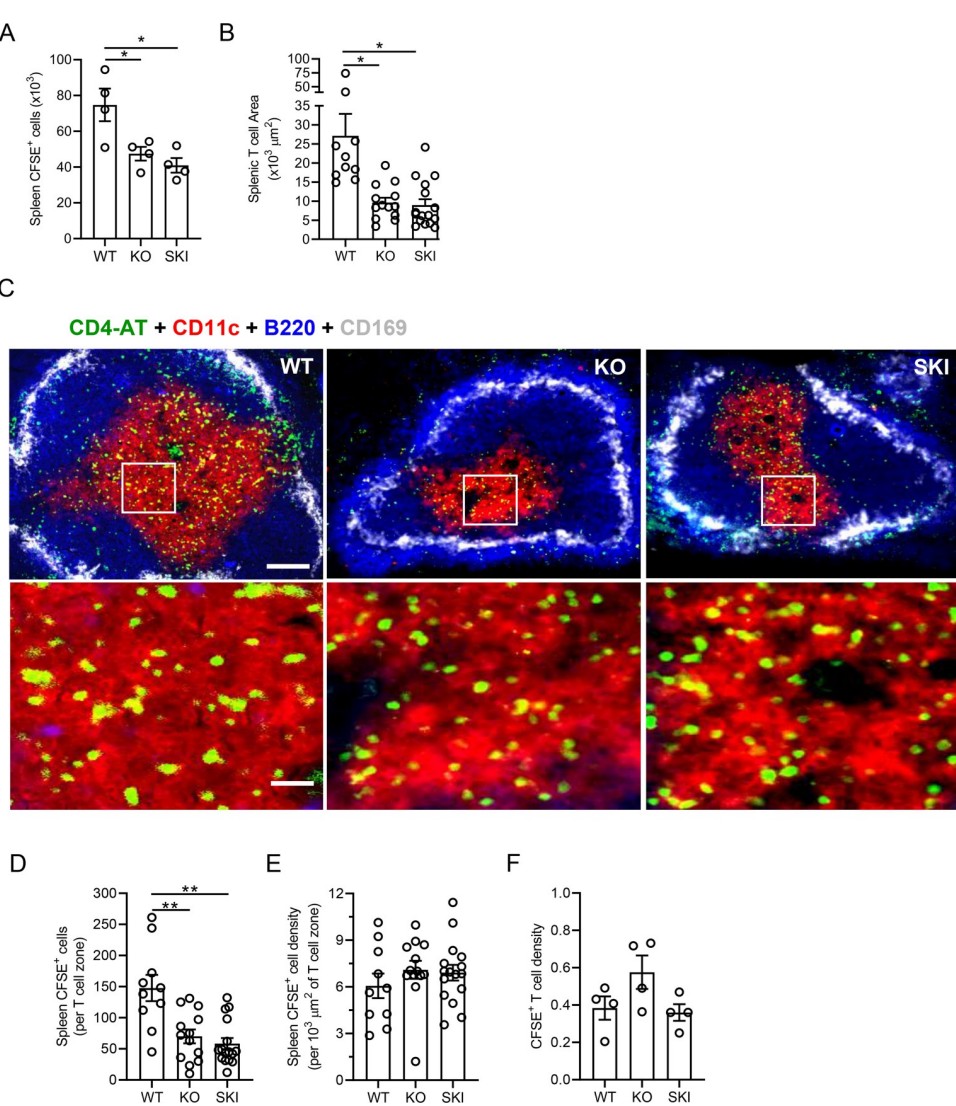

**Fig 4. OTII T cells home to and achieve similar densities in T cell zones of WT, *Cd47*KO and SKI mice. (A)** Dye-labeled WT OTII T cells were injected i.p. into WT, SKI and *Cd47*KO mice and 24 h later spleens were harvested, and the number of transferred OTII cells was determined. Mean ± SEM from 4 mice per group. **(B)** Areas of T cell zone within the white pulp of the spleens from WT, *Cd47*KO and SKI mice. Means ± SEM, n = 10–16 white pulp areas: 3 mice per group. * p < 0.05 by one-way ANOVA with Tukey test. **(C)** Immunofluorescence images of splenic white pulp of WT, *Cd47*KO and SKI mice demonstrating transferred CFSE CD4⁺ OTII T cells (green), CD4⁺ T cells (red), follicle containing B220⁺ B cells (blue) and CD169⁺ marginal zone metallophilic macrophages (white). Top images scale bar, 100 µm, 10x objective; bottom zoomed in images, scale bar, 25 µm, using a 20x objective. **(D)** Manual count of CFSE⁺ OTII T cells per T cell zone. Means ± SEM from 10–16 splenic white pulps; 3 mice per group. ** p < 0.01 by one-way ANOVA with Tukey test. **(E)** Density of CFSE⁺ OTII T cells [CFSE⁺ OTII T cells ÷ total T cell area = CFSE⁺ per 1000 µm²]. Means ± SEM from 10–16 splenic white pulp sections; 3 mice per group. **(F)** Ratio of transferred CFSE⁺ OTII T cells to endogenous CD4⁺ T cells by flow cytometric analysis. Mean ± SEM, n = 4 mice per group.

had no significant inhibitory effect while mAb to LFA-1 or ICAM-1 alone, or combined, reduced conjugate formation.

Next, we assessed Ag induced T cell expansion focusing on the function of DCs. Naïve WT OTII T cells were cocultured with BMDC from WT, *Cd47*KO or SKI mice and OVA for 48 hrs. As shown in Fig 3E, there was no difference in OTII T cell effector generation across a dose-response of OVA for *Cd47*KO and SKI OTII T cells. These results suggest that the CD47

–SIRPα axis is required for steps in T cell responses that are not modelled in this in vitro assay. Hence, we conducted further in vivo studies to investigate the cause of poor T cell priming in vivo.

## SKI and *Cd47*KO naïve T cells homing to spleen is not altered despite a reduced size of the splenic T cell zone

Next, we asked whether transferred WT OTII T cells homed into the spleens of *Cd47*KO and SKI to the same degree as WT mice under the conditions employed for adoptive transfer studies reported in Figs 1 and 2. We observed a significant reduction in WT OTII T cell numbers recovered after 24 h from spleens of *Cd47*KO (35% reduction; p<0.05) and SKI (43%; p<0.05) recipients as compared to WT mice (Fig 4A). While significant apoptosis was observed at 72 h in PBS and Ova immunized *Cd47*KO and SKI (S1A and S1B Fig), we suspected the reduction in cells was more likely explained by a reduced T cell zone size in the spleens of *Cd47*KO and SKI mice. To address this possibility, 4-color immunofluorescence microscopy was used to identify CFSE transferred OTII cells (green) within the T cell zone (red). mAb B220 was used as a B cell zone marker (blue) and a mAb specific for CD169, which is expressed by marginal zone metallophilic macrophages (white) was used to define the white pulp marginal zone landmark. ImageJ software was used to quantify the area of T cell zones in fixed and stained splenic tissues. We used this approach because reliable identification of OTII cells in the white pulp of spleen by quantitative intravital imaging was not technically possible. T cell zones in *Cd47*KO and SKI mice were often segmented and significantly reduced in area compared to those of WT mice (Fig 4B and 4C top panel), confirming an earlier report [26]. Interestingly, zoomed-in images of T cell zones of WT, *Cd47*KO and SKI mice (Fig 4C, lower panels) suggested the density of CFSE labeled OTII T cells was similar in the three mice strains. Indeed, manual counts of OTII T cells per circumscribed T cell zone (Fig 4D) divided by the average area of T cell zones (Fig 4B) revealed no statistical differences in the densities of OTII cells in splenic T cell zones of WT, *Cd47*KO and SKI mice (Fig 4E). This observation was confirmed by a second approach. We calculated the ratio of CSFE$^+$ to CFSE$^-$ endogenous T cells in spleens of WT, *Cd47*KO and SKI using flow cytometry data and the CD4$^+$ T cell counts. Despite the reduction in the CD4$^+$ T cell number and T cell zone size in *Cd47*KO and SKI mice, no statistically significant differences in OTII T cell densities were observed (Fig 4F). We conclude that there is normal T cell homing, and a similar density of transferred OTII T cells per splenic area in WT, *Cd47*KO and SKI mice despite the reduced area size in *Cd47*KO and SKI mice.

## BMDC from *Cd47*KO or SKI mice exhibit defects in migration to LN in vivo

Previous studies have shown that DCs from *Cd47*KO mice were impaired in migration into the lymphatics and trafficking by lymphatic vessels to spleen and LNs [22, 23]. A different study reported DC from SKI mice had normal trafficking to LNs [13]. To address this discrepancy in migratory function between *Cd47*KO and SKI DCs we performed pilot studies and found that the migration of in vitro derived WT DCs to spleen in WT mice was too low to be reliably measured. Instead, we assessed migration to inguinal lymph nodes (LN). BMDC from WT, SKI and *Cd47*KO mice were labeled with a distinctly different fluorescent dye, mixed together in equal proportions, and adoptively transferred into the hock of WT animals [38]. Very few *Cd47*KO BMDC migrated from the site of injection to the inguinal LNs, which is consistent with the literature [22, 23] (Fig 5A), and SKI BMDC migration to the inguinal LNs

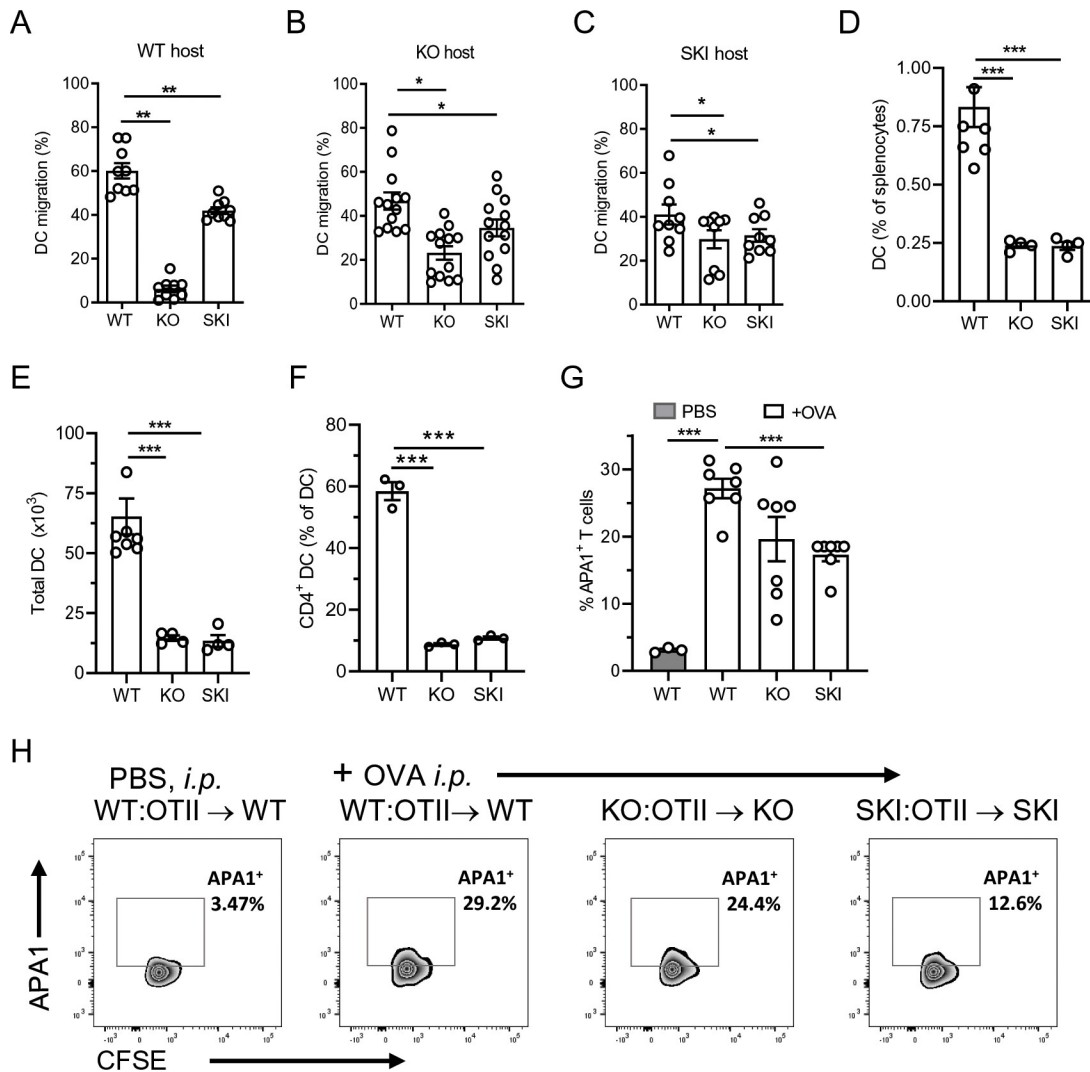

**Fig 5. SIRPα - CD47 axis regulates phenotype and function of DCs. (A-C)** Hock injected BMDC from *Cd47*KO and SKI, compared to WT mice, exhibited reduced migration. Data are expressed as the % recovery of labeled live DC (% of total recovered). Data are means ± SEM from 3 independent studies with 3–4 mice per group per study. ** p <0.01 by one-way ANOVA. **(D, E)** % DC and total DC number in spleens were reduced in *Cd47*KO and SKI compared to WT mice. **(F)** CD4+ DCs as % of total spleen DCs. **(G)** Expression of APA1/1 TCR activation epitope in CD4+ WT OTII cells recovered from spleen of WT, *Cd47*KO and SKI mice. Means ± SEM from 2 independent studies from 3–4 mice per group. *** p< 0.001 by one-way ANOVA with Tukey test. **(H)** Representative flow cytometry histograms of APA1 staining in WT, SKI and KO mice after OVA immunization.

also was significantly reduced compared to WT BMDC. Because transferred *Cd47*KO DCs may have been susceptible to clearance due to lack of CD47, a marker of self and "don't eat me signal", we examined BMDC migration in *Cd47*KO and SKI recipient mice. Here we reproduced the finding of reduced migration of BMDCs from *Cd47*KO and SKI mice in both *Cd47*KO and SKI recipients versus WT control recipients (Fig 5B and 5C).

In addition, we also reproduced prior reports that SKI and *Cd47*KO mice, compared to WT mice, have dramatically reduced total numbers of splenic DCs. This reduction is due to the dramatic reduction of CD4+ conventional CD11c+ SIRP+ DC subset (Fig 5D and 5F and S3 Fig) that are central for priming of CD4+ T cells [39, 40].

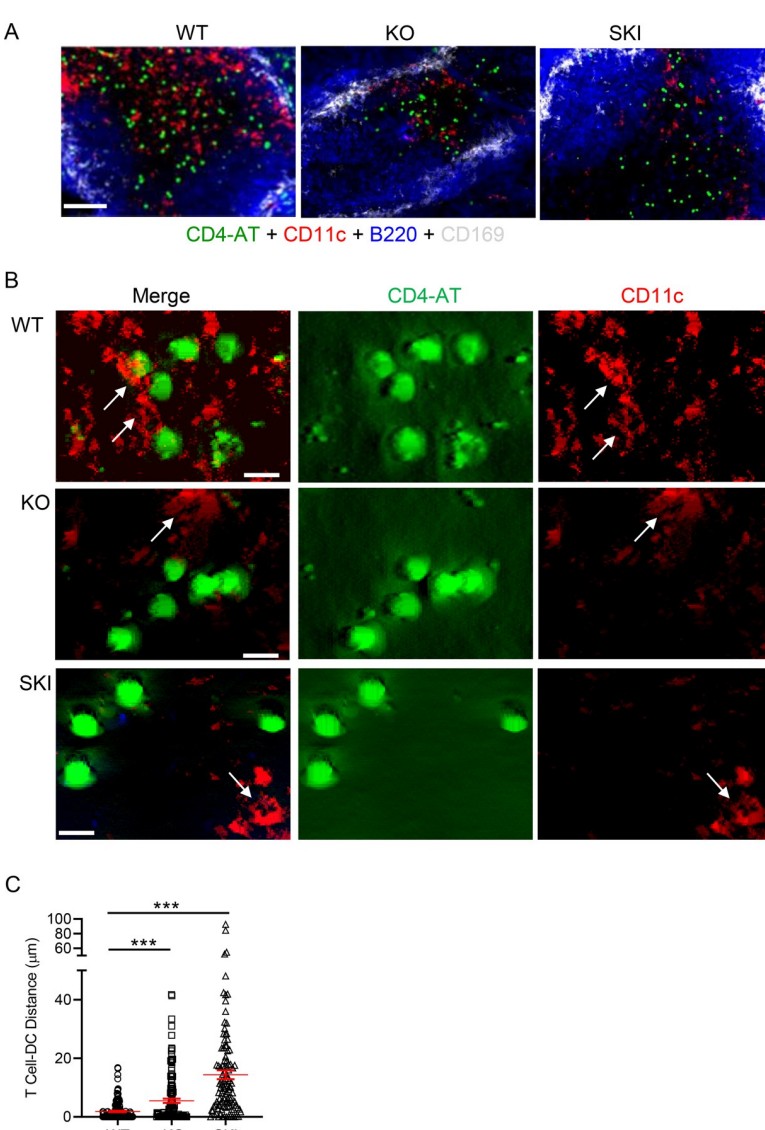

**Fig 6. Reduced CD4⁺ OTII T cell proximity to CD11c⁺ DCs in *Cd47*KO and SKI splenic T cell zones. (A)** CFSE-labeled WT OTII T cells ($4x10^6$) were i.v. injected into *Cd47*KO, SKI and WT recipient mice. Twenty-four hrs later, spleens were harvested and processed into frozen blocks. Four-color immunofluorescence microscopy of transferred CFSE⁺ OTII T cells, CD11c⁺, B cell follicles and CD169⁺ marginal zone macrophages within the splenic white pulp. Data are representative of 3–5 replicate sections from 3 mice per group. Calibration bar, 100 μm. **(B)** Immunofluorescence microscopy of CFSE⁺ T cells (green) and CD11c⁺ DCs (red) within T cell zone. Arrows identify CD11c⁺ DCs. Representative of 3 mice per group. Scale bars, (A) 25 μm and (B) 5 μm. (**C**) Proximity of CFSE CD4⁺ OTII T cells (green) to splenic CD11c⁺ (red) DCs in T cell zones of splenic white pulp in mice immunized with OVA protein. The number of CFSE⁺ T cells in T cell zone analyzed were WT, 154; *Cd47*KO, 135; SKI, 109. *** p < 0.0001 by one-way ANOVA with Welch's correction.

## Reduced TCR activation in splenic T cell zones of *Cd47*KO and SKI mice

Given the reduced number of splenic CD4⁺CD11c⁺ cDCs and the deficiencies in BMDC migration in vivo, we wondered whether Ag induced OTII activation in *Cd47*KO and SKI mice was comparable to WT mice. A prior study had reported that a MHC-antigen peptide bound to the TCR induced a conformational change exposing an "activation epitope" detected by mAb APA1/1 that binds to an epitope within the cytoplasmic tail of CD3ε [33]. APA1/1

mAb staining was readily detected in WT OTII T cells recovered from spleens 6 h after Ova immunization of WT mice compared to unimmunized mice. In contrast APA1/1 mAb staining was significantly reduced in OTII T cells recovered from immunized SKI mice (38% reduction) compared to WT. APA1/1 staining also was reduced to a similar amount in *Cd47*KO (30% reduction), but the difference did not reach statistical significance (p = 0.06) (Fig 5G). Representative flow cytometry panels are shown in Fig 5H. These data indicate that SKI and *Cd47*KO mice have defects in TCR activation due to reduced DCs—T cell interactions in the spleen.

## T cell distance to CD11c DC in splenic T cell zone is significantly greater in *Cd47*KO and SKI mice compared to WT

T cells are in constant motion guided by fibroblastic reticular network established by FRC within the splenic T cell zone. Ag recognition leads to arrest of T cells and stabilized T cell contact between relevant Ag expressing DCs. We hypothesized that defects in FRC impair formation of proper scaffold structures that create a reticular network of collagen rich matrix that bind CCL21 and CCL19 chemokines in splenic T cell zones, and together with reduced cDCs numbers and their impaired migration, lead to reduced T cell–DC proximity. To quantify DCs —T cell interactions in the splenic T cell zone in *Cd47*KO and SKI mice, we applied four-color epifluorescence microscopy and IMARIS masking software to measure the spatial distances between OTII CFSE$^+$ T cells and CD11c$^+$DCs located within T cell zones. The T cell zone in WT mice had abundant CD11c$^+$DCs and CFSE$^+$ T cells as shown in the representative micrograph (Fig 6A). In contrast, representative images of the splenic T cell zones from *Cd47*KO and SKI mice revealed obvious reductions in CFSE$^+$ T cells and CD11c$^+$ DC densities. Examination of the zoomed-in areas of micrographs of the T cell zone of WT mice showed transferred OTII T cells (CD4-AT) were in close proximity to CD11c$^+$ DC, however this was not the case in *Cd47*KO and SKI mice, which were significantly distanced from CD11c$^+$ DCs, with SKI DCs at the greatest distances as determined by quantitative immunofluorescence microscopy and IMARIS image analysis (Fig 6B and 6C and S4A–S4C Fig).

## Discussion

The SIRPα - CD47 axis plays an important role in immune cell homeostasis and in leukocyte trafficking in inflammatory and autoimmune disease models. *Cd47*KO and SKI mice have multiple defects in splenic immune cells that reduce their susceptibility to multiple inflammatory disease models. Their respective roles in Ag driven T effector cell generation, however, are incompletely defined in cellular and molecular terms. Here we demonstrate that both CD47 and SIRPα play critical roles in T cell-DC interactions within the splenic T cell zone necessary for Ag priming of naïve CD4$^+$ T cells.

## The CD47—SIRPα axis regulates DC and T cell survival and migration negatively impacting Ag priming

DC are highly efficient Ag presenting cells that initiate primary responses through activation of naïve T cells in cell mediated immunity. Our findings indicate that the reduced expansion of CD4$^+$ T cell observed in *Cd47*KO and SKI mice upon Ag immunization was primarily due to multiple defects in CD4$^+$ cDCs and an increase in apoptosis of CD4$^+$ T cells both at baseline and after Ag induced activation. These data support the conclusion that the dominant defect in both *Cd47*KO and SKI mice is the "double hit" of the dramatic reduction (>80%) in the number of splenic CD4$^+$ cDCs and their impaired migration in vivo. Previous studies have

identified defects in splenic FRC cells that we speculate leads to altered T cells and DC motion on fibroreticular networks and contribute to reduced interactions between DCs and OTII T cells. Functionally, these multiple defects are the likely cause of the reduced proximity of OTII to DCs in splenic T cell zones, leading to significant reductions in Ag dependent TCR activation in *Cd47*KO and SKI mice (Figs 5G and 6). We also propose that defects in T cell adhesive function can contribute to the impaired Ag priming in *Cd47*KO and SKI mice as exemplified by our prior study in *Cd47*KO CD4[+] T cells [5, 15] and the data in S2 Fig that identify defects in LFA-1 integrin adhesive function in vivo and in vitro. Thus, CD47 was initially identified as integrin associated protein or IAP [41]. As that name implies, CD47 has been shown to regulate the activation and adhesive function of LFA-1 and VLA-4 integrins in human and mouse CD4[+] T cells [5, 15] and CD11b/CD18 (Mac-1) integrin in human and mouse neutrophils [6]. Hence, we speculate that the migration defect in *Cd47KO* MDDC is related to defective integrin adhesive function. On the other hand, it is currently unknown how defective signaling by SIRPα leads to defects in DC migration and requires further study. An unexpected finding was that SKI OT-II T cells did not recover after transfer into WT hosts as compared to WT T cells (Fig 2G). Since CD4[+] T cells had been reported to express little if any SIRPα, it was unknown how the impaired signaling of SIRPα in SKI mice led to defective adhesive and migratory functions of CD4 T cells. Our analysis of SKI T cell function revealed defects in LFA-1 adhesive function (S2 Fig). Future studies are necessary to elucidate the underlying mechanisms by which impaired SIRPα signaling leads to defects in LFA-1 adhesive functions in T cells.

Prior studies found splenic CD4[+] T cells were reduced by 30–40% in both *Cd47*KO and SKI mice due to reductions in FRC that provide secretion of IL-7, a T cell survival cytokine [42] and CCL19 and CCL21 chemokines, which are crucial to promote both CD4[+] T cell and DC recruitment [26, 28]. While the current experimental design did not detect major defects in vitro in Ova induced expansion of OTII T cells prepared from *Cd47*KO or SKI mice, our previous study had reported that activation of *Cd47*KO CD4[+] T cells resulted in delayed apoptosis after more than 72 h of culture as detected by Annexin V and TUNEL staining, which was not related to impaired expression of the anti-apoptotic protein Bcl-xL [15]. In the current study CD4[+] OTII T cells from both *Cd47*KO and SKI mice exhibited elevated apoptosis compared to WT OTII T cells at baseline and after Ag immunization with a time frame of 48 to 72 h after transfer. We did not find evidence supporting an important role for splenic clearance of adoptively transferred OTII cells as factors contributing to poor Ag T cell activation. There is a significant literature for both CD47 and SIRPα participation in apoptotic cell death in DCs, eosinophils, and platelets [25, 43, 44].

### *Cd47*KO and SKI exhibited impaired Ag induced DC-T cell interaction and TCR activation in vivo

An important observation was that loss of the SIRPα-CD47 axis signaling impaired the TCR activation step required for Ag priming. While prior studies have inferred this outcome, here we assessed this question directly in vivo. Previous studies have shown that the APA1/1 mAb detects murine T cell TCR activation upon Ag immunization in vivo and in in vitro conditions [32, 33]. Using the OTII TCR Tg mice and the APA1/1 mAb we directly demonstrated that adoptively transferred WT naïve OTII T cells have impaired activation in *Cd47*KO and SKI recipient mice compared to WT mice. This reduction in TCR activation was readily detected in the spleen.

### Impact of diminished FRC in T cell zones on T cell priming

The FRC create supportive highway networks and instructive niches for immune cells in secondary lymphoid organs that enable adaptive immune responses (reviewed in [45, 46]). FRC

secrete CCL19 and CCL21 chemokines and IL7 and form scaffold structures that create a reticular network of collagen rich matrix in splenic T cell zones that facilitates efficient migration of DCs and T cells leading to meaningful interactions, and act as conduits for distribution of antigens and CCL21 and CCL19 chemokines [47, 48]. In particular in vitro studies have reported that CCL21 associates with reticular fibers to promote DC haplotaxis and that CCL19 diffuses and promotes chemotaxis [48]. Furthermore, CCL19 has been shown to promote dendrite extension in splenic DCs [49]. Thus, we propose that the previously reported reduction of splenic FRC reticular network and production of IL7 and chemokines CCL21 and CCL19 in *Cd47*KO and SKI mice also contribute to the reduced Ag induced T cell priming [24, 26].

## Conclusion

Our study extends our previous studies that the observed lack of T cell activation and expansion in *Cd47*KO and SKI mice is caused by the reduced number of CD4$^+$ DCs in the spleen caused by their reduced migration into lymphoid organs associated with defective and reduced numbers of FRC, and by defects in the survival and migration of CD4$^+$ T cells in vivo. Modulation of the CD47-SIRPα axis may be useful for tuning T cell responses for therapeutic purposes.

## Supporting information

**S1 Fig. Annexin V staining analysis of transferred OTII T cells isolated from spleens.** (A,B) WT, *Cd47*KO (KO) and SKI naive OTII T cells were labeled with violet dye and transferred i. p. into WT, KO and SKI recipient mice, respectively. After 24 hr mice were injected i.p. with OVA (A) or PBS (B) and spleens harvested after 24, 48 or 72 h. CD4$^+$ OTII T cells were assessed for Annexin V and PI staining by flow cytometry.
(PDF)

**S2 Fig. Splenocytes from WT and SKI mice were stained with primary labeled mAb to CD4, LFA-1, VLA-4, CD47 and SIRPα and primary labeled isotype-matched control mAb and fluorescence was detected by flow cytometry and analyzed by FlowJo software.** Data are representative of 3 WT and SKI mice. **(B)** T cell arrest on ICAM-1 coimmobilized with 200 ng/100 μL of SDF-1α (CXCL12) at an estimated laminar shear stress of 1.0 dynes/cm$^2$. n = 2 separate experiments, replicates performed in triplicate. $^*$ p<0.05 by Student t test. **(C-E)** WT and SKI mice had increased immune cell infiltrates in response to rmTNF-α in a dermal air pouch model of recruitment. SKI mice had significantly reduced influx of CD4$^+$ T cells and neutrophils compared to WT mice. N = 3–5 mice per group in 2 separate experiments $^*$p<0.05, $^{**}$p<0.01, $^{***}$ P< 0.005.
(PDF)

**S3 Fig. Numbers of dendritic cells (DC), especially CD4$^+$ DCs, are decreased in the spleens of *Cd47*KO and SKI mice compared to WT mice.** Representative plots of CD45$^+$, CD11c$^+$ MHCII$^{hi}$ DC and CD4$^+$ DC in WT (62.3%), *Cd47*KO (9.39%) and SKI (10.7%) mice as determined by flow cytometry.
(PDF)

**S4 Fig. Detection of CD4$^+$ CFSE-labeled T cell distance from CD11c$^+$ splenic DCs by IMARIS software. A–C.** Representative three-color images of splenic sections were imported into IMARIS software, surface renderings were performed to identify B cell zones (blue cells) surrounding T cells zones that contain transferred CD4$^+$ OTII T cells (green) and endogenous CD11c stained DCs (red). Computation of distances of DCs from CFSE cells were determined by IMARIS from 11 *Cd47*KO, 11 SKI and 8 WT mouse samples. Calibration bars = 50μm.

Data are representative images of different T cell zones in spleen analyzed for 3 mice per group.
(PDF)

## Acknowledgments

We thank members of our Center for Excellence in Vascular Biology for their helpful suggestions and discussions. We thank Dr. Jerry Turner, BWH, for assistance in fluorescence microscopy and providing the use of IMARIS software for image analysis. AA, HW, FVP and GN performed experiments, analyzed results, and prepared figures; AA, HW, GN, DE and FWL designed studies; PE and CAP provided critical reagents; HW, AA, and FWL wrote the manuscript; DE and AHL provided critical review of the manuscript and figures.

## Author Contributions

**Conceptualization:** Gail Newton, Charles A. Parkos, Pablo Engel, Daniel Engelbertsen, Andrew H. Lichtman, Francis W. Luscinskas.

**Data curation:** Huan Wang, Francisco Velázquez, Gail Newton, Daniel Engelbertsen, Francis W. Luscinskas.

**Formal analysis:** Anu Autio, Huan Wang, Francisco Velázquez, Gail Newton, Daniel Engelbertsen, Francis W. Luscinskas.

**Funding acquisition:** Francis W. Luscinskas.

**Investigation:** Anu Autio, Huan Wang, Francisco Velázquez, Gail Newton, Daniel Engelbertsen, Francis W. Luscinskas.

**Methodology:** Anu Autio, Huan Wang, Gail Newton, Daniel Engelbertsen, Andrew H. Lichtman.

**Project administration:** Anu Autio, Gail Newton, Francis W. Luscinskas.

**Resources:** Charles A. Parkos, Pablo Engel.

**Supervision:** Gail Newton, Francis W. Luscinskas.

**Writing – original draft:** Francis W. Luscinskas.

**Writing – review & editing:** Anu Autio, Huan Wang, Gail Newton, Charles A. Parkos, Daniel Engelbertsen, Andrew H. Lichtman.

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
