## [Decision Letter · Decision Letter 0]

27 Jan 2022

PONE-D-21-40921SIRPα - CD47 interactions regulate dendritic cell-T cell interactions and TCR activation during T cell priming in spleenPLOS ONE

Dear Dr.  Luscinskas

Thank you for submitting your manuscript to PLOS ONE. After careful consideration, we feel that it has merit but does not fully meet PLOS ONE’s publication criteria as it currently stands. Therefore, we invite you to submit a revised version of the manuscript that addresses the points raised during the review process.

We look forward to receiving your revised manuscript.

Kind regards,

Anshu Agarwal

Academic Editor

PLOS ONE

Journal Requirements:

2. PLOS requires an ORCID iD for the corresponding author in Editorial Manager on papers submitted after December 6th, 2016. Please ensure that you have an ORCID iD and that it is validated in Editorial Manager. To do this, go to ‘Update my Information’ (in the upper left-hand corner of the main menu), and click on the Fetch/Validate link next to the ORCID field. This will take you to the ORCID site and allow you to create a new iD or authenticate a pre-existing iD in Editorial Manager. Please see the following video for instructions on linking an ORCID iD to your Editorial Manager account: https://www.youtube.com/watch?v=_xcclfuvtxQ.

Reviewers' comments:

Reviewer's Responses to Questions

**Comments to the Author**

1. Is the manuscript technically sound, and do the data support the conclusions?

Reviewer #1: Yes

Reviewer #2: Partly

2. Has the statistical analysis been performed appropriately and rigorously? 

Reviewer #1: Yes

Reviewer #2: Yes

3. Have the authors made all data underlying the findings in their manuscript fully available?

Reviewer #1: Yes

Reviewer #2: Yes

4. Is the manuscript presented in an intelligible fashion and written in standard English?

Reviewer #1: Yes

Reviewer #2: Yes

5. Review Comments to the Author

Reviewer #1: Autio A and Wang H et al., report on the role of the SIRPa-CD47 axis in regulating DC-T cell interactions. Well-written manuscript extending the team’s previous work on CD47. Using several genetic mice tools, authors show that CD47 is necessary for CD4 T cell activation and proliferation in vivo. Because DC-T cell interactions in vitro are no affected and the homing of T cells to the secondary lymphoid tissues is also similar, authors test the hypothesis that SIRPa-CD47 is required for the migration of activated DCs carrying the antigens do the secondary lymphoid organs. This is shown using multiplexed BMDC migration assay in vivo and quantifying DC numbers in the spleen.

Major comments: Greater T-DC distance in CD47KO and SKI mice may be because of decreased number of DCs in the spleen due to migration defects as demonstrated in figure 5. Experimental details on data in figure 6 need more explanation. Figure 6A legend reads “Four-color immunofluorescence microscopy of transferred CFSE+ OTII T cells, CD11c+, B cell follicles and CD169+ marginal zone macrophages within the splenic white pulp”. How were the cells were injected i.v, i.p, or hock?

Minor comment: Image quality is lost in transformation.

Reviewer #2: In the manuscript by Luscinskas et al., the authors focused on the role of the CD47-SIRPα axis in the antigen-specific DC-T interaction and demonstrated impaired OT-II T cell proliferation in the spleen of CD47 KO or SIRPα KI mice after OVA immunization. The authors also focused on the interaction of T cells and DCs in the spleen and provided the results suggesting the reduced TCR activation in OT-II T cells transplanted into CD47 KO and SIRPα KI mice, as well as the reduced proximity of DCs to T cells in the spleen of these mice. These results suggest the importance of the interaction between CD47 and SIRPα in the context of the antigen-specific immune response, however, some of the data are still immature and inappropriately concluded. Thus, some modification is necessary for the publication to PloS One.

Major comments

1. Figure 5: Since the spleen does not contain lymphatics, it remains unclear whether the reduction of DC migration from skin to the draining lymph node through lymphatics is related to the reduction of DCs in the spleen. In addition, the previous study suggested that the cause of the reduced splenic CD4+ cDCs in CD47 KO or SIRPα KI mice is still unclear and is in part implicated in the activation of the subset, particularly in the upregulation of CCR7, which can promote the migration of DCs (Yi et al. Immunity 2015: PMID 26453377). Thus, there is not enough evidence to support the reduction of splenic CD4+ cDCs is attributable to the impaired migration of BMDCs from CD47KO and SIRPα KI mice, and the claim “dual effect” described in the discussion is not clear from these results. To show the importance of DC migration in these mice, the author should at least demonstrate if the results shown in Figure 5A-C are dependent on the CCR7-CCL19/21 axis or the activation of BMDCs.

2. As most of the data demonstrated in the present study are related to published studies, the data shown in Figure 6G is worthwhile to indicate the presence of the study. Therefore, the authors should precisely show the data with representative FACS plot of APA1+ T cells in each group together with isotype-matched control staining.

3. In Figure 7, since the number of DC and CD4 T cells is decreased in the T cell zone of CD47 KO or SIRPα KI mice, the distance between them must increase. Thus, this reviewer cannot understand the necessity to measure the distance of these cells in this Figure. To claim the importance of the CD47-SIRPα axis on DC-T proximity, the authors should demonstrate 1) whether the prolonged distance between T cells and DCs is independent of the reduction in the number of these cells in the spleen of CD47 KO or SIRPα KI mice, as well as 2) whether the proximity of these cells is dependent on transferred OVA antigen, particularly in the control mice.

Minor comments

1. “SIRPα-CD47 interactions” in the manuscript title is an overstatement since the authors did not show direct evidence showing the importance of the interaction between CD47 and SIRPα.

2. In Figure 2G, the authors should explain or discuss the reason why SKI OT-II T cells did not recover in WT hosts, although T cells are absent for SIRPα.

3. Regarding the DC-T cell conjugate formation assay shown in Figure 3, the authors should indicate when OVA is supplemented during the culture.

4. The legend in Figure 4E is described in 4F and should be replaced with each other.

5. Nishimura et al. (PMID 32438469) recently demonstrated that the differentiation of effector T cells is attenuated in the draining LN of MOG-immunized DC-specific SIRPα deficient mice. The paper could be cited to mention the importance of SIRPα on DCs for the T cell priming.

6. PLOS authors have the option to publish the peer review history of their article (what does this mean?). If published, this will include your full peer review and any attached files.

Reviewer #1: **Yes: **Shivashankar Othy

Reviewer #2: No

---

## [Author Response · Author response to Decision Letter 0]

9 Mar 2022

We appreciate the Reviewers helpful comments and suggestions to improve the work and data presentations. We have addressed the Reviewers comments by including new results and revising the Abstract, Introduction, Results and Discussion sections and appropriate Figures.

---

## [Decision Letter · Decision Letter 1]

23 Mar 2022

SIRPα - CD47 axis regulates dendritic cell-T cell interactions and TCR activation during T cell priming in spleen

PONE-D-21-40921R1

Dear Dr. Luscinskas,

We’re pleased to inform you that your manuscript has been judged scientifically suitable for publication and will be formally accepted for publication once it meets all outstanding technical requirements.

Kind regards,

Anshu Agarwal

Academic Editor

PLOS ONE

Additional Editor Comments (optional):

T

Reviewers' comments:

Reviewer's Responses to Questions

**Comments to the Author**

1. If the authors have adequately addressed your comments raised in a previous round of review and you feel that this manuscript is now acceptable for publication, you may indicate that here to bypass the “Comments to the Author” section, enter your conflict of interest statement in the “Confidential to Editor” section, and submit your "Accept" recommendation.

Reviewer #2: All comments have been addressed

2. Is the manuscript technically sound, and do the data support the conclusions?

Reviewer #2: Yes

3. Has the statistical analysis been performed appropriately and rigorously? 

Reviewer #2: Yes

4. Have the authors made all data underlying the findings in their manuscript fully available?

Reviewer #2: Yes

5. Is the manuscript presented in an intelligible fashion and written in standard English?

Reviewer #2: Yes

6. Review Comments to the Author

Reviewer #2: In the revised manuscript by Autio et al., the authors replied to our comments with providing new data and rewrote the manuscript. Basically, it is well described, but the following issues have remained.

Minor comments

1. Figure 5H seems to be difficult to see the differences between each group in the frequency of APA1-positive cells shown by the pseudocolor plot. It would be better to other way to show it (e.g. contour or zebra plot with larger dots).

2. Page 23, line 500; FCR seems like a misspelling of FRC.

7. PLOS authors have the option to publish the peer review history of their article (what does this mean?). If published, this will include your full peer review and any attached files.

Reviewer #2: No

---

## [Editor Report · Acceptance letter]

1 Apr 2022

PONE-D-21-40921R1 

SIRPa - CD47 axis regulates dendritic cell-T cell interactions and TCR activation during T cell priming in spleen 

Dear Dr. Luscinskas:

I'm pleased to inform you that your manuscript has been deemed suitable for publication in PLOS ONE. Congratulations! Your manuscript is now with our production department. 

Kind regards, 

on behalf of

Dr. Anshu Agarwal 

Academic Editor

PLOS ONE